# Improving Visual Grounding with Pixel-Word Correlation and Cross-layer Regularization

## Abstract

Visual grounding aims to localize the target object in an input image according to a language expression. To achieve this purpose, existing methods either extract the visual and linguistic features independently or utilize language information to guide visual feature extraction. However, the former strategy generates identical, general-purpose visual representations for different text queries, which are redundant and sub-optimal for visual grounding tasks. Other methods that adopt the latter scheme typically construct sophisticated modules based on linguistic features, implicitly guiding the visual feature extraction through end-to-end training. But they often overlook fine-grained visual-linguistic alignment information, resulting in less discriminative visual features, and thus limiting overall model performance. In this paper, we propose a simple-yet-effective module named Highlighter, which explicitly calculates pixel-word correlations between visual and linguistic features, and then uses this correlation information to calibrate and enhance the visual representations. Based on the proposed module, we further introduce a cross-layer regularization loss, designed to maintain the consistency of fine-grained alignment information across different layers and to facilitate the transmission of supervision signals to shallow layers of the visual encoder. Extensive experiments demonstrate that our method achieves state-of-the-art performance on five widely used visual grounding datasets. And ablation studies also verify the effectiveness and efficiency of our method.

## 1 Introduction

Visual grounding is a general object detection task (Yao et al., 2024a), which aims to localize the target object or region in an input image that is most relevant to a natural language description. Due to its potential in bridging the gap between visual perception and language expression, visual grounding has emerged as a central problem in many multi-modal reasoning researches (Kamath et al., 2021; Li et al., 2022; Zou et al., 2023; Zhang et al., 2022).

To achieve the above purpose, inspired by DETR (Carion et al., 2020) based detectors, the previous methods like TransVG (Deng et al., 2021) and MDETR (Kamath et al., 2021) re-formulate the visual grounding as an end-to-end coordinates regression problem. They typically design a two-branch network by leveraging the transformer architecture (Vaswani et al., 2017), in which two parallel encoders are employed to extract the visual and linguistic features from the input images and language expressions, respectively. However, a major problem with this parallel-encoder design is that the extraction processes of visual and linguistic features are completely independent. As a result, given an input image with multiple objects, the visual encoder will output identical, general-purpose feature representations for different text queries (Ye et al., 2022). But actually, the multi-modal fusion decoder only requires the foreground visual features that associated with the input language expression to localize the target object. This inconsistency will decrease their performance in solving the visual grounding task.

Some recently proposed works have noticed this problem and introduced various language-guided visual encoder structures to deal with it. These methods can be roughly divided into two categories: **(1) The feature-based methods** directly manipulate the intermediate visual features based on the language information (Ye et al., 2022; Yang et al., 2022a) or integrate such information into the visual representations through the cross-attention mechanism (Yang et al., 2022b; Deng et al., 2023).

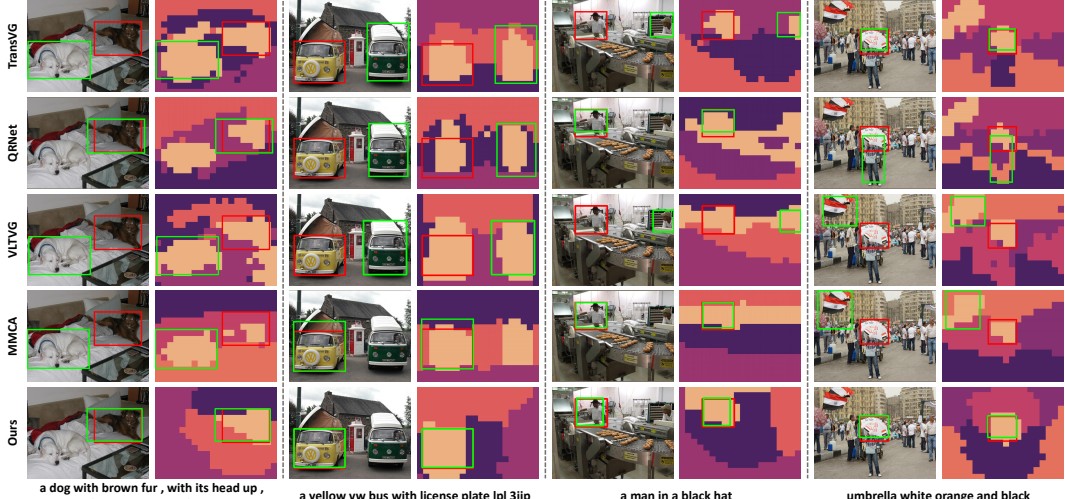

Figure 1: K-means clustering comparison between different state-of-the-art methods and our framework. By exploiting the pixel-word correlation information, our Highlighter module can accurately distinguish the pixels of target object and other image regions. The model predictions and ground-truth annotations are indicated by the green and red bounding boxes, respectively.

**(2) The structure-based methods** dynamically modify the parameters (Su et al., 2023b; Luo et al., 2024; Yao et al., 2024b) or the architecture (Su et al., 2023a; Uzkent et al., 2023) of the visual encoder according to the linguistic features, such that the language-relevant visual features can be extracted. Although performance gains can be achieved with the enhanced visual features, these methods typically require a sophisticated design, such as query-aware attention module (Ye et al., 2022), dynamic weight generator (Su et al., 2023b; Yao et al., 2024b) and gumbel-softmax-based gating mechanism (Su et al., 2023a), which may increase the difficulty of model training.

And moreover, we also find that most of the above mentioned methods cannot accurately distinguish the target object from those belonging to the same category. To show this, we perform K-means clustering on different pixels of the visual features extracted by TransVG (Deng et al., 2021), QRNet (Ye et al., 2022), VLTVG (Yang et al., 2022a) and MMCA (Yao et al., 2024b), respectively. As can be seen from the results in Figure 1, for all these four methods, there are many pixels outside the target region (i.e., red bounding box) are grouped into the same cluster with the text-referred object. These pixels will confuse the models and make them incorrectly detect another object of the same category as the target (see predicted green bounding box). We analyze the main reason for such problem is that these methods do not explicitly align the visual and linguistic features between local pixels and each word. Consequently, they may overlook certain descriptive words specific to the target object, and thus indiscriminately highlight some other image regions associated with similar semantic concepts. This will reduce the discriminability of the extracted visual features and eventually decrease the detection accuracy.

To alleviate these problems, we propose the Highlighter, a simple-yet-effective module which *explicitly calculates a correlation map between visual and linguistic features at the pixel-word level, and then utilizes it to spatially calibrate and enhance the visual representations*. Specifically, the Highlighter first projects the visual features produced at each intermediate encoder layer and the extracted linguistic features into a common space to align their dimensions. The matrix multiplication operation is then used to calculate the pixel-word correlation map between these two projected embeddings. After that, we generate word-wise weights based on the linguistic features to perform attentive pooling over the correlation values, such that the pixel-wise responses of the visual features can be obtained. Finally, the Highlighter spatially recalibrates the original visual features according to their corresponding responses, and feeds the calibrated representations back into the visual encoder to make it more focused on language-relevant image regions. As shown in Figure 1, by exploiting fine-grained pixel and word alignment information, our Highlighter can more accurately partition the pixels of the target object and other image regions into different clusters,

thereby improving the discriminability of its calibrated visual features. In addition, to ensure that the same input text will highlight similar areas of the visual features in different encoder layers, we further introduce a cross-layer regularization loss, which utilizes the ground-truth to supervise the spatial responses generated by each Highlighter module. Specifically, it converts the bounding box annotations into binary spatial masks, and then minimizes the KL (Kullback-Leibler) divergence between these masks and the corresponding pixel-wise response maps. By doing so, this loss can also effectively transmit the supervision signals to shallow layers of the visual encoder, thus facilitating the training process of model parameters. Extensive experiments on the RefCOCO, Ref-COCO+, RefCOCOg, ReferItGame, and Flickr30k Entities datasets show that our method achieves new state-of-the-art performance, demonstrating its superiority in solving the visual grounding task. Ablation studies also validate the effectiveness and efficiency of our method. In summary, our main contributions are three-folds:

(1) We propose a Highlighter module, which explicitly calculates and exploits the fine-grained pixel-word correlation information to improve the visual feature extraction process, such that more attention can be paid to language-relevant image regions. We also introduce a cross-layer regularization loss to promote the consistency of focused areas across the Highlighter modules in different layers.

(2) By integrating the proposed Highlighter modules, we establish a visual grounding framework. Since the visual representations extracted by our encoder are sufficiently effective, further feature enhancement is not required during the decoding phase. This allows our framework to utilize a more computationally efficient decoder for final predictions.

(3) Extensive experiments and ablation studies are conducted on five widely used datasets to evaluate the performance and efficiency of our method. The results indicate that our method can promote the visual encoder to better focus on the region referred to by the input expression and significantly improve the detection accuracy.

## 2 RELATED WORK

**Visual Grounding.** Early visual grounding methods typically extend the general object detection framework and can be roughly divided into two categories: (1) Two-stage approaches (Yang et al., 2019a; Yu et al., 2018; Zhang et al., 2018; Liu et al., 2019a; Chen et al., 2021; Hong et al., 2019) first utilize a pre-trained detector to generate a set of object proposals, and then select the most text-relevant one by matching them with the input referring expression. (2) One-stage methods (Chen et al., 2018; Yang et al., 2019b; Liao et al., 2020; Qiu et al., 2020; Yang et al., 2020) make a dense prediction at each spatial position of a feature map, which is generated by fusing the visual and linguistic features.

Recently, with the success of Vision Transformer (Dosovitskiy et al., 2020; Vaswani et al., 2017) in object detection and vision-language tasks, a series of transformer-based visual grounding models have been proposed. Referring Transformer (Li & Sigal, 2021) proposes to generate contextualized lingual queries from image-text joint embeddings, and produces detection and segmentation predictions based on these queries. TransVG (Deng et al., 2021) reformulates the visual grounding as a coordinates regression problem. It feeds a learnable query token along with visual and linguistic features into a multi-modal fusion module, and uses it for target coordinates prediction. MDETR (Kamath et al., 2021) tackles visual grounding tasks through a text-modulated detection framework derived from the DETR detector (Carion et al., 2020). Dynamic MDETR (Shi et al., 2023) further introduces a 2D adaptive sampling module to select more informative image patches, which reduces the spatial redundancy and speeds up the visual grounding process. Although improved performance have been achieved, the visual feature extraction process of these methods are completely independent from the language expression, which will eventually limit their effectiveness in solving the visual grounding task.

**Language-guided Visual Encoder.** Several approaches have noticed the above issue and attempt to deal with it by designing language-guided visual encoders. Both QRNet (Ye et al., 2022) and LAVT (Yang et al., 2022b) utilize the attention mechanism to realize early fusion of visual and linguistic features at intermediate layers of the vision transformer encoder. VLTVG (Yang et al., 2022a) incorporates a visual-linguistic verification module to explicitly model the relationships between visual and linguistic features. LG-FPN(Suo et al., 2022) further performs the attention operation on image

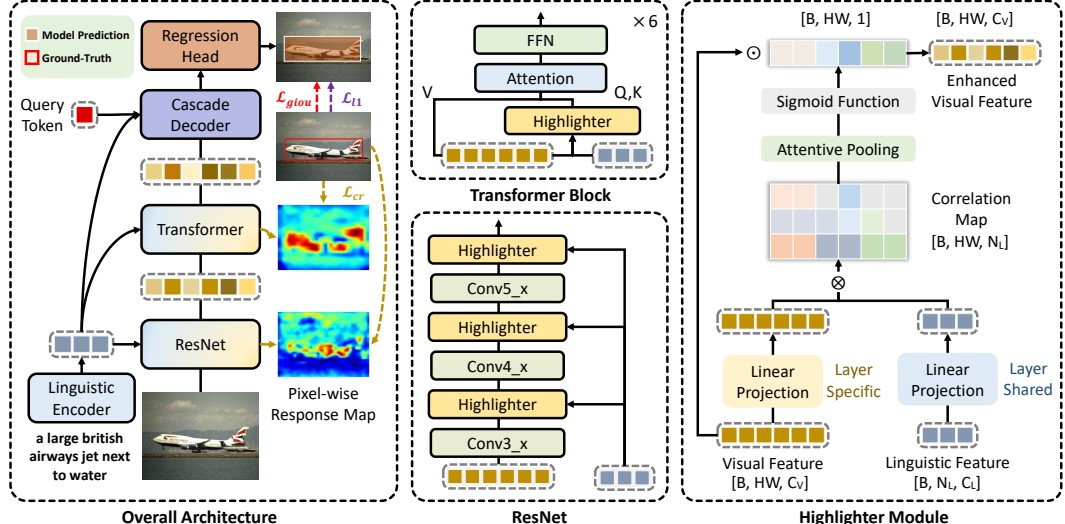

Figure 2: Overview of our adopted end-to-end framework. It utilizes a visual encoder (implemented as a ResNet followed by 6 transformer blocks) and a linguistic encoder for feature extraction, and employs a multi-stage cascade decoder to generate the final predictions. We illustrate the workflow of our proposed Highlighter module and also show how to introduce it into the ResNet and Transformer sub-networks.

features at different scales to achieve language-guided cross-scaled information fusion. TransVG++ (Deng et al., 2023) proposes a Language Conditioned Vision Transformer, which integrates the linguistic features into visual tokens via a language prompter or a language adapter. VG-LAW (Su et al., 2023b) constructs an expression-specific visual backbone by generating its dynamic weights according to the input referring expression. LADS (Su et al., 2023a) uses binary gates to adaptively select sub-networks from the entire visual encoder based on the referring expressions. However, most of these methods rely on some sophisticated and complex designs, which may ignore some visual-linguistic alignment information and result in sub-optimal feature extraction.

## 3 METHODOLOGY

### 3.1 ARCHITECTURE OVERVIEW

In this section, we present the overall architecture of our framework, which is established following a typical end-to-end encoder-decoder paradigm (Deng et al., 2021; Ye et al., 2022), as illustrated in Figure 2. More specifically, the encoder part of our framework utilizes a visual encoder and a linguistic encoder to extract the corresponding features from the input images and language expressions, respectively. The proposed Highlighter module is hierarchically embedded into the visual encoder to make the extracted features more focused on language-relevant image regions, promoting its effectiveness in the visual grounding task. For the decoder part, a multi-stage cascade decoder is employed. It introduces a learnable query token (`[REG]` token) as an additional input and then performs the cross-attention on it with the linguistic and visual features alternatively, such that the language-relevant target object information can be effectively aggregated. Finally, similar to (Yang et al., 2022a), based on the output representations of `[REG]` token, a regression head predicts the bounding box coordinates $\hat{b}_i = (\hat{x}_i, \hat{y}_i, \hat{w}_i, \hat{h}_i)$ of the text-referred object at each decoder stage. Based on these predictions, the training loss for optimizing our framework is formulated as:

$$\mathcal{L} = \sum_{i=1}^{L} [\lambda_{L1}\mathcal{L}_{L1}(\hat{b}_i, b) + \lambda_{giou}\mathcal{L}_{giou}(\hat{b}_i, b)] + \lambda_{cr}\mathcal{L}_{cr}, \tag{1}$$

where $\mathcal{L}_{L1}(\cdot, \cdot)$ and $\mathcal{L}_{giou}(\cdot, \cdot)$ represent the smooth L1 loss (Girshick, 2015) and the GIoU loss (Rezatofighi et al., 2019), respectively. $\mathcal{L}_{cr}$ indicates our introduced cross-layer response regularization loss, which will be described in detail in Section 3.4. $\lambda_{L1}$, $\lambda_{giou}$ and $\lambda_{rc}$ are three positive

parameters that balance the relative importance of these losses. $L$ denotes the total number of decoder stages and $b = (x, y, w, h)$ represents the ground-truth bounding box. After training, only $\hat{b}_L$ in the last stage is taken as the prediction result for the testing data.

## 3.2 THE VISUAL AND LINGUISTIC ENCODERS

For feature extraction, we build the visual and linguistic encoders as follows:

**Visual Encoder:** Following the setup in previous works (Deng et al., 2021; Yang et al., 2022a), we adopt a ResNet network (He et al., 2016) (e.g., ResNet-50 or ResNet-101) followed by 6 transformer layers as the visual encoder. Given an input image, we flatten its feature map produced by the ResNet along the spatial dimension. The resulted features are then added with the positional embeddings and fed into the following transformer layers to generate the final visual representations.

**Linguistic Encoder:** We employ a basic BERT (Devlin et al., 2018) network with 12 layers as the linguistic encoder. For each input language expression, we process the sentence text as in (Deng et al., 2021; Yang et al., 2022a; Ye et al., 2022) and feed it into the BERT to extract its linguistic features $F_l \in \mathbb{R}^{N_l \times C_l}$. Here, $N_l$ is the length of the input expression and $C_l$ denotes the feature dimension of each word token. As our main objective is to highlight the language-relevant regions in the visual features, we do not introduce any specific design for this encoder.

## 3.3 THE PROPOSED HIGHLIGHTER MODULE

To accurately localize the referred target object in an input image according to a natural language expression, it is better for the extracted visual features to focus on the local regions that are relevant to the text description. To this end, we propose a simple-yet-effective Highlighter module, which enhances the visual representations by utilizing a correlation map calculated between visual and linguistic features at the pixel-word level, as illustrated in Figure 2. Specifically, given the flattened features $F_v \in \mathbb{R}^{HW \times C_v}$ extracted by one intermediate layer of the visual encoder and the linguistic features $F_l \in \mathbb{R}^{N_l \times C_l}$ of an input sentence, the Highlighter module $\mathcal{F}_{\text{HL}}(F_v, F_l)$ first projects them into a common space for dimension alignment, and then calculates their correlation map as follows:

$$U = F_v W_v (F_l W_l)^{\text{T}}. \tag{2}$$

$W_v \in \mathbb{R}^{C_v \times C_s}$ and $W_l \in \mathbb{R}^{C_l \times C_s}$ are two linear projectors, which convert the feature dimension of each pixel in $F_v$ (i.e., $C_v$) and that of each word token in $F_l$ (i.e., $C_l$) into the same dimension value $C_s$, respectively. Here, $W_l$ is shared across all Highlighter modules to ensure that the correlation calculations in different encoder layers can be performed in an identical semantic space. With this definition, the obtained correlation map $U \in \mathbb{R}^{HW \times N_l}$ can model the pixel-word relationships between $F_v$ and $F_l$, where $H$, $W$ indicate the height and width of the visual feature map.

Generally, different words in each input expression have varying importance in referring to the target object. For example, conceptual nouns and related descriptive words are often more informative and important than others. With this consideration, we propose to generate word-wise attention weights based on $F_l$ to indicate the informativeness of each word token as follows:

$$\alpha_i = \frac{\exp(\mathcal{F}_{\text{P}}(F_l[i]))}{\sum_{i=1}^{N_l} \exp(\mathcal{F}_{\text{P}}(F_l[i]))}, \tag{3}$$

where $\mathcal{F}_{\text{P}}(\cdot)$ denotes a non-linear projection function that maps the linguistic feature $F_l[i]$ of the $i$-th word token to its corresponding attention weight $\alpha_i$. $\mathcal{F}_{\text{P}}(\cdot)$ consists of two linear layers with a layer normalization (LN) operation and a GeLU non-linear activation between them. According to the generated weights, we perform the following attentive pooling along the word dimension of $U$ to obtain the pixel-wise responses of $F_v$ for the whole sentence:

$$\widetilde{U} = \sigma(\sum_{i=1}^{N_l} \alpha_i U[i]). \tag{4}$$

Here, $U[i]$ represents the $i$-th column of $U$ and $\sigma(\cdot)$ denotes the sigmoid function that normalizes each element value of $\widetilde{U}$ into the range of $(0, 1)$. With the above operation, $\widetilde{U} \in \mathbb{R}^{HW \times 1}$ aggregates the correlation information between each pixel feature and all word tokens. The larger element value in $\widetilde{U}$ indicates the stronger text relevance of the corresponding pixel. Therefore, we can consider $\widetilde{U}$ as a spatial weight mask and apply it to $F_v$ for feature calibration:

$$\mathcal{F}_{\text{HL}}(F_v, F_l) = \widetilde{U} \circ F_v, \tag{5}$$

where $\circ$ is a broadcasting element-wise multiplication operation, which makes the calibrated features $\widetilde{U} \circ F_v$ pay more attention to the language-relevant pixels. This will eventually improve the effectiveness of the visual representations produced by our Highlighter module $\mathcal{F}_{\mathrm{HL}}(F_v, F_l)$ for solving the visual grounding task.

**Applied to Visual Encoder:** As shown in Figure 2, the Highlighter module described above can be easily integrated into both the convolutional and transformer layers. For the ResNet part of our visual encoder, we introduce this Highlighter module after the Conv3_x, Conv4_x and Conv5_x layers as:

$$\widetilde{F}_v = \mathcal{F}_{\mathrm{HL}}(F_v, F_l) + F_v. \tag{6}$$

After that, the enhanced features $\widetilde{F}_v \in \mathbb{R}^{HW \times C_v}$ will be reshaped to the size of $H \times W \times C_v$ and then fed into the next layer. Note that the values of $H$, $W$ and $C_v$ can be different for the feature maps that extracted from different layers.

And for each transformer encoder block, the Highlighter module is inserted before the linear projection of query and key tokens in the attention layer, intended to highlight language-relevant regions by adjusting the attention scores, which can be formulated as:

$$q, k = \mathcal{F}_{\mathrm{HL}}(F_v, F_l), v = F_v, \tag{7}$$

$$\widetilde{F}_v = \mathcal{F}_{\mathrm{MHSA}}(q, k, v) + F_v, \tag{8}$$

where $\mathcal{F}_{\mathrm{MHSA}}(\cdot, \cdot, \cdot)$ denotes the multi-head self-attention layer. The output $\widetilde{F}_v$ that is enhanced by $\mathcal{F}_{\mathrm{HL}}(F_v, F_l)$ will be taken as the input to the subsequent LN layer and feed-forward network.

### 3.4 THE CROSS-LAYER REGULARIZATION LOSS

The visual features $F_v$ extracted by the different encoder layers typically vary considerably in terms of receptive field size, degree of information abstraction and some other factors. Hence, the response matrix $\widetilde{U}$ that is calculated from $F_v$ can also vary greatly across Highlighter modules. However, for the same input expression, it is expected to highlight similar spatial areas for different features $F_v$. To explicitly achieve this consistency, we propose the cross-layer regularization loss $\mathcal{L}_{cr}$, which applies distribution constrains to $\widetilde{U}$ generated by each layer's Highlighter module. Specifically, we first convert the ground-truth annotation $b = (x, y, w, h)$ into a binary mask $M$ of the same size as the input image, where the pixel values inside the bounding box $b$ are set to 1 and others are set to 0. Then, we resize this mask $M$ so that it has an identical spatial resolution with $\widetilde{U}$ of different Highlighter modules. Finally, we spatially normalize the flattened $\widetilde{U} \in \mathbb{R}^{HW}$ and $M \in \mathbb{R}^{HW}$ by the softmax operation, and define our loss $\mathcal{L}_{cr}$ as the KL divergence between them:

$$M'[i] = \frac{\exp(M[i]/\tau)}{\sum_{i=1}^{HW} \exp(M[i]/\tau)}, U'[i] = \frac{\exp(\widetilde{U}[i]/\tau)}{\sum_{i=1}^{HW} \exp(\widetilde{U}[i]/\tau)}, \tag{9}$$

$$\mathcal{L}_{cr} = \sum_{n=1}^{N_h} \sum_{i=1}^{HW} M'[i] \cdot \log(\frac{M'[i]}{U'[i]}), \tag{10}$$

where $M[i]$ denotes the $i$-th element in matrix $M$. $\tau$ is a temperature parameter that controls the distribution shape of $M'$ and $U'$. We empirically set it to 0.2 in all our experiments. $N_h$ indicate the total number of integrated Highlighter modules. Optimizing $\mathcal{L}_{cr}$ will minimize the distribution differences between $M$ and $\widetilde{U}$. This can make most of the large values in each $\widetilde{U}$ concentrate within the target areas indicated by $M$, so that the text-referred regions in different $F_v$ can all be highlighted correctly. Additionally, with the above operations, the supervision signals in ground-truth $b$ can be more effectively transmitted to shallow layers of the visual encoder through the binary mask $M$. This may further benefit the training process of model parameters.

### 3.5 THE CASCADE DECODER

After feature encoding, most previous methods continually refine and fuse the visual and linguistic features in the decoding procedure. In contrast, by integrating the Highlighter modules, our encoder can output more effective visual representations, in which the language-relevant regions are already

highlighted. Therefore, similar to (Yang et al., 2022a; Liu et al., 2023), we adopt a cascade decoder that utilizes a learnable query token (`[REG]`) to iteratively aggregate useful information from the extracted visual and linguistic features. This process removes the computational overhead for further feature enhancement and thus increases the efficiency of our whole framework, as validated in Section 4.3. The output embedding of `[REG]` token is then passed to a regression head to generate the final prediction. **More details of the cascade decoder are shown in Appendix A.1.**

## 4 EXPERIMENTS

### 4.1 IMPLEMENTATION DETAILS

**Datasets.** To evaluate the detection performance and computational efficiency of our entire framework, as well as to validate the effectiveness of our proposed Highlighter module, we conduct extensive experiments and a series of ablation studies on five commonly used datasets, including RefCOCO (Yu et al., 2016), RefCOCO+ (Yu et al., 2016), RefCOCOg (Mao et al., 2016), Refer-ItGame (Kazemzadeh et al., 2014) and Flickr30k Entities (Plummer et al., 2015). **We provide a more detailed introduction of these datasets in Appendix A.2.**

For all these datasets, we resize each image sample to make its longer edge equal to 640 and then pad its shorter edge to 640 as well. So all the input images in our experiments have a spatial size of $640 \times 640$. We also set the maximum length of each language expression to 40 and truncate the words that exceed this length. The special `[CLS]` and `[SEP]` tokens are appended to the beginning and end of each expression before it is input into the linguistic encoder.

**Training Details.** To train our proposed framework, we employ the AdamW optimizer (Loshchilov & Hutter, 2017) with batch size of 64 and weight decay of $10^{-4}$. Similar to (Deng et al., 2023; Yang et al., 2022a), our visual and linguistic encoders are initialized by well pre-trained DETR (Carion et al., 2020) and BERT (Devlin et al., 2018) models, respectively, and then optimized with an initial learning rate of $10^{-5}$. As for our cascade decoder, we randomly initialize its parameters by the Xavier (Glorot & Bengio, 2010) scheme and set its initial learning rate to $10^{-4}$. To make a fair comparison with other methods, we similarly train our model for 90 epochs and decrease the learning rate by a factor of 10 after 60 epochs. In addition, we also adopt the data augmentation strategy used in the previous works (Yang et al., 2019b; 2020; Deng et al., 2021; Ye et al., 2022) during the training phase. The common space dimension value $C_s$ is set to $C_s = 256$. For the loss function in Equation (1), the hyperparameters are as $\lambda_{L1} = 5, \lambda_{giou} = 2, \lambda_{cr} = 1$ and the number of decoder stages $L$ is set to $L = 6$.

**Experimental Environment.** All experiments in this section are performed on a server with an Intel(R) Xeon(R) Gold 6133 @2.50GHz CPU and the Ubuntu 20.04.4 LTS operating system. Our model is trained by using two NVIDIA A100 Tensor Core GPUs (32 samples per GPU for each mini-batch with a size of 64) and tested on a single NVIDIA GeForce RTX 3090 GPU.

**Evaluation Metric.** We follow the standard ACC@0.5 protocol (Yang et al., 2019b; 2020; Deng et al., 2021; Ye et al., 2022) to evaluate the detection performance of our framework for the visual grounding task. Specifically, given an image-expression pair as input, the predicted bounding box is considered as correct only if the IoU value calculated between it and the corresponding ground-truth target is greater than 0.5.

### 4.2 COMPARISON WITH STATE-OF-THE-ART METHODS

**RefCOCO/RefCOCO+/RefCOCOg.** Table 1 summarizes the detection accuracy of our framework and current state-of-the-art visual grounding methods (including both one-stage and two-stage approaches) on the RefCOCO, RefCOCO+ and RefCOCOg datasets. From these experimental results, we can obtain the following observations:

(1) Our proposed framework significantly outperforms all two-stage methods. One of the main reasons for this phenomenon is that the performance of the two-stage methods is heavily dependent on a pre-trained object detector. But both the visual features extracted by this detector and its generated object proposals may be not suitable for the visual grounding task, thus finally leading to incorrect predictions.

Table 1: Performance comparison with state-of-the-art methods on the RefCOCO, RefCOCO+ and RefCOCOg datasets. We highlight the best and second best results obtained with the same backbone in **bold** and underlined.

| Methods | Backbone | RefCOCO | | | RefCOCO+ | | | RefCOCOg | | |
|---|---|---|---|---|---|---|---|---|---|---|
| | | val | testA | testB | val | testA | testB | val-g | val-u | test-u |
| **Two-stage** | | | | | | | | | | |
| MAttNet (Yu et al., 2018) | ResNet-101 | 76.65 | 81.14 | 69.99 | 65.33 | 71.62 | 56.02 | - | 66.58 | 67.27 |
| RvG-Tree (Hong et al., 2019) | ResNet-101 | 75.06 | 78.61 | 69.85 | 63.51 | 67.45 | 56.66 | - | 66.95 | 66.51 |
| CM-A-E (Liu et al., 2019b) | ResNet-101 | 78.35 | 83.14 | 71.32 | 68.09 | 73.65 | 58.03 | - | 67.99 | 68.67 |
| NMTree (Liu et al., 2019a) | ResNet-101 | 76.41 | 81.21 | 70.09 | 66.46 | 72.02 | 57.52 | 64.62 | 65.87 | 66.44 |
| Ref-NMS (Chen et al., 2021) | ResNet-101 | 80.70 | 84.00 | 76.04 | 68.25 | 73.68 | 59.42 | - | 70.55 | 70.62 |
| PBREC-MT(Zhao et al., 2024) | ResNet-101 | 82.94 | 86.31 | 80.81 | 74.85 | 79.53 | 65.60 | - | 73.86 | 74.13 |
| **One-stage** | | | | | | | | | | |
| ReSC-L(Yang et al., 2020) | DarkNet-53 | 77.63 | 80.45 | 72.30 | 63.59 | 68.36 | 56.81 | 63.12 | 67.30 | 67.20 |
| SAFF (Ye et al., 2021) | DarkNet-53 | 79.26 | 81.09 | 76.55 | 64.43 | 68.46 | 58.43 | - | 68.94 | 68.91 |
| TransVG (Deng et al., 2021) | ResNet-50 | 80.32 | 82.67 | 78.12 | 63.50 | 68.15 | 55.63 | 66.56 | 67.66 | 67.44 |
| D-MDETR (Shi et al., 2023) | ResNet-50 | 81.62 | 83.85 | 76.24 | 67.00 | 70.95 | 58.13 | 68.04 | 70.14 | 69.57 |
| LADS (Su et al., 2023a) | ResNet-50 | 82.85 | 86.67 | 78.57 | 71.16 | 77.64 | 59.82 | - | 71.56 | 71.66 |
| VLTVG (Yang et al., 2022a) | ResNet-50 | 84.53 | 87.69 | 79.22 | 73.60 | 78.37 | 64.53 | 72.53 | 74.90 | 73.88 |
| MMCA (Yao et al., 2024b) | ResNet-50 | 84.34 | 86.99 | 80.06 | 72.44 | 78.01 | 63.86 | 72.02 | 74.11 | 73.46 |
| Our Method | ResNet-50 | **85.00** | **87.89** | **80.64** | **74.24** | **79.85** | **64.89** | **74.24** | **76.98** | **76.54** |
| TransVG (Deng et al., 2021) | ResNet-101 | 81.02 | 82.72 | 78.35 | 64.82 | 70.70 | 56.94 | 67.02 | 68.67 | 67.73 |
| LG-FPN (Suo et al., 2022) | ResNet-101 | 81.76 | 84.78 | 78.16 | 70.29 | 76.19 | 59.68 | 69.20 | 73.06 | 73.24 |
| LUNA (Liang et al., 2023) | ResNet-101 | 84.67 | 86.74 | 80.21 | 72.79 | 77.98 | 64.61 | - | 74.16 | 72.85 |
| VLTVG (Yang et al., 2022a) | ResNet-101 | 84.77 | 87.24 | 80.49 | 74.19 | 78.93 | 65.17 | 72.98 | 76.04 | 74.18 |
| MMCA (Yao et al., 2024b) | ResNet-101 | 84.76 | 87.34 | 80.86 | 73.18 | 78.67 | 64.13 | 72.53 | 74.91 | 73.87 |
| Our Method | ResNet-101 | **85.10** | **88.43** | **81.50** | **75.42** | **80.74** | **65.39** | **74.35** | **77.80** | **77.26** |
| TransVG++(Deng et al., 2023) | ViT-S | 85.24 | 87.50 | 80.46 | 73.73 | 79.21 | 63.56 | 73.43 | 74.78 | 74.77 |
| PVD(Cheng et al., 2024) | Swin-B | 84.99 | 88.02 | 80.03 | 74.27 | 79.06 | 65.11 | 74.34 | 74.64 | 71.41 |
| QRNet(Ye et al., 2022) | Swin-S | 84.01 | 85.85 | 82.34 | 72.94 | 76.17 | 63.81 | 71.89 | 73.03 | 72.52 |
| VG-LAW(Su et al., 2023b) | Swin-S | 84.82 | 87.22 | 81.94 | 74.36 | 78.49 | 65.24 | - | 75.61 | **76.28** |
| Our Method | Swin-S | **85.93** | **88.51** | **82.65** | **74.88** | **80.22** | **65.47** | **74.71** | **75.97** | 75.73 |

Figure 3: Visualization results of VLTVG and our method. The predictions and ground-truth targets are indicated by the green and red bounding boxes, respectively.

(2) Compared with the one-stage approaches, our framework generally outperforms the previous best VLTVG method on the RefCOCO dataset in most settings, and consistently produces the highest detection accuracy on the RefCOCO+ and RefCOCOg datasets. This shows that enhancing the visual representations through our Highlighter module to focus more on the language-relevant regions can bring great benefits to solving the visual grounding task. Moreover, our method obtains more significant performance gains on RefCOCOg than on the other two datasets, typically exceeding VLTVG by about 2.00% and outperforming other benchmark methods by more than 3.00%. This may demonstrate the advantage of utilizing the pixel-word correlation to detect target objects referred to by long language expressions.

(3) Some recently proposed methods, QRNet, VG-LAW and PVD, utilize the Swin-Transformer (Liu et al., 2021) (Swin-S and Swin-B) as their backbone network, which is more powerful than the ResNet architecture in extracting informative visual features (Deng et al., 2023; Ye et al., 2022).

Despite this, our framework with ResNet backbone still performs better than these approaches. And when the Swin-S backbone is adopted (the Highlighter modules are integrated into the first layer of every Swin-S stage), our method achieves superior performance on almost all evaluation sets of the three datasets. This also indicates the effectiveness of our Highlighter module and shows its compatibility with different backbone architectures.

In Figure 3, we further visualize the detection results produced by the previous best VLTVG (Yang et al., 2022a) method and our framework on the RefCOCOg dataset. We observe that both VLTVG and our method can successfully understand the object concept of the referred target, while our method outperforms VLTVG in capturing the visual contextual information that is associated with some descriptive words. For example, given the expression "black shirt girl", these two approaches both recognize a "girl" in the image, but only our method correctly detects the target girl wearing the "black shirt". Therefore, VLTVG may fail to process images that contain multiple objects belonging to the same class as the referred target. In contrast, our framework benefits from visual-linguistic alignment at the pixel-word level, thereby effectively handling these cases.

**ReferItGame/Flickr30k Entities.** In Table 2, we present the performance of different methods on the test sets of the ReferItGame and Flickr30K Entities datasets. Concretely, with a ResNet-50 backbone, our detection accuracy on the two datasets is 74.43% and 80.55%, which exceeds other competing approaches by 2.83% $\sim$ 4.67% on ReferItGame and 1.37% $\sim$ 2.08% on Flickr30K. When using a stronger ResNet-101 backbone, the performance of our framework can be improved to 75.71% and 81.63%, surpassing the benchmark methods by 2.74% $\sim$ 4.98% and 1.79% $\sim$ 2.53% on the ReferItGame and Flickr30K

Table 2: Performance comparison with state-of-the-art methods on the test sets of the ReferItGame and Flickr30K Entities datasets.

| Methods | Backbone | ReferItGame | Flickr30K |
|---|---|---|---|
| **One-stage** | | | |
| ReSC-L(Yang et al., 2020) | DarkNet-53 | 64.60 | 69.28 |
| TransVG (Deng et al., 2021) | ResNet-50 | 69.76 | 78.47 |
| LADS (Su et al., 2023a) | ResNet-50 | 71.08 | - |
| VLTVG (Yang et al., 2022a) | ResNet-50 | 71.60 | 79.18 |
| Our Method | ResNet-50 | **74.43** | **80.55** |
| TransVG (Deng et al., 2021) | ResNet-101 | 70.73 | 79.10 |
| LUNA (Liang et al., 2023) | ResNet-101 | 72.97 | 79.45 |
| VLTVG (Yang et al., 2022a) | ResNet-101 | 71.98 | 79.84 |
| Our Method | ResNet-101 | **75.71** | **81.63** |
| QRNet (Ye et al., 2022) | Swin-S | 74.61 | 81.95 |
| VG-LAW(Su et al., 2023b) | Swin-S | 74.83 | - |
| Our Method | Swin-S | **76.20** | **82.54** |

datasets, respectively. Furthermore, once again, our framework achieves comparable results to QR-Net and VG-LAW that use the Swin-S backbone. All the above experimental results demonstrate the superiority and generality of our method in dealing with different visual grounding tasks.

## 4.3 ABLATION STUDY

**Effect of the Highlighter Module.** In Table 3, we conduct a series of ablation experiments on the RefCOCOg dataset, to investigate the impact of our proposed Highlighter module in terms of visual grounding performance, model size (number of parameters), computational cost (GFLOPs), inference time for each image-expression pair, and GPU memory usage for model training. Specifically, we integrate the Highlighter modules into two simple baseline methods, namely TransVG and TransVG (Cas)(Deng et al., 2021), to compare the above metrics with and without our module. Here, TransVG (Cas) is established by replacing the vision-language fusion module in TransVG with the cascade decoder described in Section 3.5. From the results, it can be observed that:

(1) For both baselines, integrating our Highlighter modules can improve their accuracy by a large margin. Concretely, TransVG+Highlighter outperforms the original TransVG by 6.46% $\sim$ 6.91% and such improvements increase to 9.10% $\sim$ 9.53% for TransVG (Cas), suggesting that these models can benefit from the visual representations enhanced by the Highlighter modules. Note that even the poorly performing TransVG+Highlighter can still surpass the recently proposed LADS (Su et al., 2023a), LUNA (Liang et al., 2023) and QRNet (Ye et al., 2022) methods (see Table 1).

(2) Compared with TransVG, by using the cascade decoder, TransVG (Cas) consistently introduces only 0.82M additional parameters, but has lower computational overhead (39.51/69.85 GFLOPs vs. 41.16/71.49 GFLOPs) and less inference time (8.53/11.92 ms vs. 8.80/12.33 ms). More importantly, TransVG (Cas) significantly reduces the training GPU memory costs of TransVG by 22.61%

Table 3: Performance and efficiency comparison of different baselines on the RefCOCOg dataset with and without our Highlighter module.

| Methods | Backbone | RefCOCOg | | Params | GFLOPs | Inference Time | GPU Memory |
|---|---|---|---|---|---|---|---|
| | | val-u | test-u | | | | |
| TransVG | ResNet-50 | 67.66 | 67.44 | 149.52M | 42.35 | 8.95 | 26.1G |
| +Highlighter | ResNet-50 | 74.37 (+6.71) | 74.00 (+6.56) | 151.62M | 43.99 | 9.22 | 27.5G |
| TransVG | ResNet-101 | 68.67 | 67.73 | 168.46M | 72.61 | 12.67 | 31.4G |
| +Highlighter | ResNet-101 | 75.13 (+6.46) | 74.64 (+6.91) | 170.56M | 74.26 | 13.07 | 32.8G |
| TransVG (Cas) | ResNet-50 | 67.88 | 67.01 | 150.34M | 39.51 | 8.53 | 20.2G |
| +Highlighter | ResNet-50 | 76.98 (+9.10) | 76.54 (+9.53) | 152.44M | 41.16 | 8.80 | 21.5G |
| TransVG (Cas) | ResNet-101 | 68.44 | 67.89 | 169.28M | 69.85 | 11.92 | 26.8G |
| +Highlighter | ResNet-101 | 77.80 (+9.36) | 77.26 (+9.37) | 171.38M | 71.49 | 12.33 | 28.4G |

Table 4: Performance and efficiency comparison of introducing our Highlighter module in different sub-networks of the visual encoder.

| ResNet | Transformer | $\mathcal{L}_{cr}$ Loss | RefCOCOg | | Params | GFLOPs | Inference Time |
|---|---|---|---|---|---|---|---|
| | | | val-u | test-u | | | |
| - | - | - | 67.88 | 67.01 | 150.34M | 39.51 | 8.53 |
| ✓ | - | - | 75.76 (+7.88) | 75.41 (+8.40) | 151.65M | 40.99 | 8.69 |
| - | ✓ | - | 75.17 (+7.26) | 74.61 (+7.60) | 151.33M | 39.68 | 8.64 |
| ✓ | ✓ | - | 76.16 (+8.28) | 75.92 (+8.91) | 152.44M | 41.16 | 8.80 |
| ✓ | ✓ | ✓ | **76.98 (+9.10)** | **76.54 (+9.53)** | 152.44M | 41.16 | 8.80 |

(26.1G→20.2G) and 14.65% (31.4G→26.8G), respectively. Therefore, the cascade decoder we adopt is more computationally efficient than the vision-language fusion module used in TransVG.

(3) TransVG (Cas)+Highlighter performs better than TransVG+Highlighter, which indicates that our proposed module is more compatible with the cascade decoder by calibrating the visual features, demonstrating the overall effectiveness of our framework.

**Where to Integrate the Highlighter Module.** To explore where the Highlighter module should be introduced, we separately integrate it into the ResNet and Transformer parts of our visual encoder. Note that if neither of these sub-networks utilize the proposed module, then our framework will degenerate into the TransVG (Cas) model. The ResNet-50 backbone is employed for this ablation study. As shown in Table 4, when equipping our Highlighter module in one of the ResNet and Transformer sub-networks, it brings 7.88%/8.40% and 7.26%/7.60% performance gains compared to the baseline, respectively. In comparison, our framework integrates the Highlighter into both sub-networks and achieves better visual grounding accuracy of 76.16%/75.94%. These results significantly outperform the baseline by 8.28%/8.91% without substantial loss in model efficiency (only introduce 1.65 extra GFLOPs and take 0.27 ms more for inference).

Additionally, introducing our proposed cross-layer regularization loss $\mathcal{L}_{cr}$ can further improve the model performance to 76.98%/76.54%, demonstrating the benefit of achieving cross-layer response consistency in accurately highlighting the text-referred regions in the visual features of different encoder layers. **Please refer to the appendix for more experimental results and discussions.**

## 5 CONCLUSION

This work proposes a Highlighter module for improving visual grounding performance. It only consists of several simple operations, but can effectively leverage the pixel-word correlation information to calibrate and enhance the visual representations. In this way, the extracted visual features will pay more attention to the language-relevant regions, so that the visual-linguistic alignment can be explicitly achieved. In addition, a multi-stage cascade decoder is also employed to further improve the effectiveness and efficiency of our framework. Experiments indicate that our method achieves new state-of-the-art performance on five datasets, surpassing recently proposed VLTVG, LADS, LUNA, QRNet, VG-LAW and MMCA.

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

# A APPENDIX

## A.1 DETAILS OF THE CASCADE DECODER

Our decoder network consists of multiple stages with the same architecture and a regression head shared by all these stages as shown in Figure 4. For information aggregation, a learnable query token ([REG]) $R_0 \in \mathbb{R}^{1 \times C_r}$ is introduced as an input, and then updated at each decoder stage through the cross-attention operations. Specifically, for the $i$-th stage ($1 \leq i \leq L$), given the enhanced visual representations $\widetilde{F}_v$ and the extracted linguistic features $F_l$, the [REG] embedding $R_{i-1} \in \mathbb{R}^{1 \times C_r}$ from the previous stage is first successively fed into two cascading multi-head attention (MHA) layers as follow:

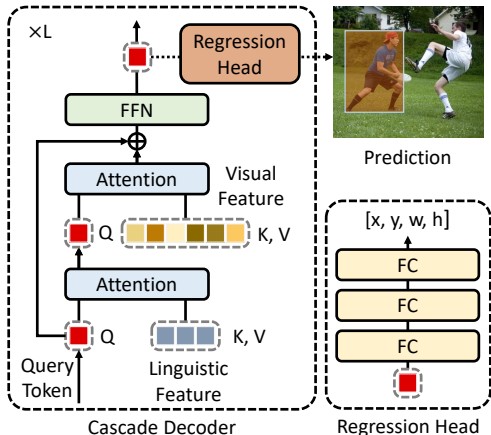

$$R_{i-1}^l = \mathcal{F}_{\mathrm{MHA}}(R_{i-1}, F_l, F_l) \qquad (11)$$

$$R_{i-1}^v = \mathcal{F}_{\mathrm{MHA}}(R_{i-1}^l, \widetilde{F}_v, \widetilde{F}_v) \qquad (12)$$

Figure 4: Illustration of the cascade decoder and regression head used in our framework.

Here, $\mathcal{F}_{\mathrm{MHA}}(\cdot, \cdot, \cdot)$ denotes the multi-head attention function, in which the three arguments represent the query, key and value inputs. According to the above formula, the generated $R_{i-1}^l$ and $R_{i-1}^v$ will summarize semantically meaningful information from $F_l$ and then collect useful visual features of the target object from $\widetilde{F}_v$, respectively. This can also be viewed as an implicit multi-modal fusion process. After that, we use $R_{i-1}^v$ to update the original $R_{i-1}$ such that the query token embedding $R_i \in \mathbb{R}^{1 \times C_r}$ for the $i$-th stage can be obtained as:

$$\widetilde{R}_{i-1} = \mathcal{F}_{\mathrm{LN}}(R_{i-1}^v + R_{i-1}), \quad R_i = \mathcal{F}_{\mathrm{LN}}(\mathcal{F}_{\mathrm{FFN}}(\widetilde{R}_{i-1}) + \widetilde{R}_{i-1}). \qquad (13)$$

Here, $\mathcal{F}_{\mathrm{LN}}(\cdot)$ indicates the layer normalization operation. $\mathcal{F}_{\mathrm{FFN}}(\cdot)$ represents a feed-forward network that composed of two linear layers with ReLU activations. The updated embedding $R_i$ is then fed into the next stage for iterative information aggregation and cross-modal reasoning.

Finally, the outputs of our cascade decoder $\{R_i\}_{i=1}^L$ are all input into a regression head to generate the corresponding bounding box predictions $\{\hat{b}_i\}_{i=1}^L = \{(\hat{x}_i, \hat{y}_i, \hat{w}_i, \hat{h}_i)\}_{i=1}^L$ (see Equation (1)). This regression head is a fully connected network containing a stack of three linear layers with ReLU activation functions.

## A.2 INTRODUCTION OF THE USED DATASETS

**RefCOCO** (Yu et al., 2016) includes 19,994 images with 50,000 referred objects, where each image may contain multiple instances from the same object categories. There are 142,210 referring expressions in total, so each instance may get more than one text description. Following the standard setup, the image samples in RefCOCO are officially split into train/validation/testA/testB subsets that have 120,624/10,834/5,657/5,095 expressions, respectively.

**RefCOCO+** (Yu et al., 2016) consists of 19,992 images with 49,856 referred objects and 141,564 referring expressions. The usage of location-related words (e.g., "left" or "right") is strictly disallowed in the expressions from this dataset. RefCOCO+ is also officially split into train, validation, testA and testB sets with 120,191, 10,758, 5,726 and 4,889 expressions, respectively.

**RefCOCOg** (Mao et al., 2016) has 95,010 long expressions collected on Amazon Mechanical Turk for 49,856 referred objects in 25,799 images. Among them, 85,474 expression-referent pairs are selected for model training, and the remaining 9,536 expressions are separated following two different strategies, namely RefCOCOg-google (Mao et al., 2016) (val-g) and RefCOCOg-umd (Nagaraja et al., 2016) (val-u and test-u). We conduct experiments on these two partitions to make comprehensive comparisons.

**ReferItGame** (Kazemzadeh et al., 2014) includes 20,000 images collected from the SAIAPR-12 dataset (Escalante et al., 2010). Following the same split as in the previous works (Deng et al., 2021; Ye et al., 2022), we construct a train set with 54,127 expressions, a validation set with another 5,842 expressions, and a test set with the remaining 60,103 expressions.

**Flickr30k Entities** (Plummer et al., 2015) is a large-scale dataset with 427k referred entities in 31,783 images. It is built based on the original Flickr30K (Young et al., 2014) dataset by utilizing region-to-phrase correspondences for image description. We follow the previous works (Yang et al., 2020; 2022a; Deng et al., 2023) to divide the image samples: 29,783 for training, 1,000 for validation, and 1,000 for testing.

### A.3 Analysis of Referring Expression Length

To analyze the impact of referring expression length on detection performance, we conducted a comparison of different expression lengths on the test-u split of RefCOCOg dataset.[1] As shown in the Figure 5, the results indicate a downward trend in detection accuracy as the length of the referring expression increases. We believe this decline is due to the challenges that longer text expressions pose to the model's multimodal reasoning capabilities. Nevertheless, our method maintains an accuracy of 76.3% for targets with referring expressions of 11 words or more. Compared to previous methods such as TransVG (Deng et al., 2021), MMCA (Yao et al., 2024b) and VLTVG (Yang et al., 2022a), Highlighter consistently demonstrates superior detection accuracy across varying expression lengths, which also reflects the effectiveness of our method in handling longer expressions.

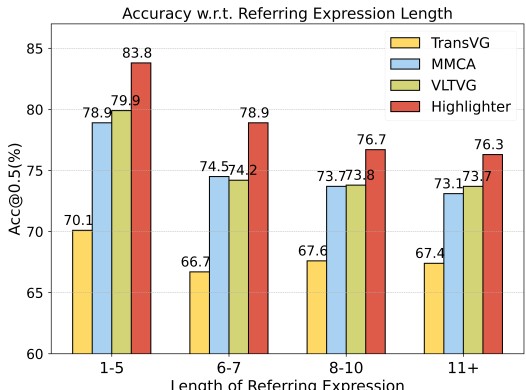

Figure 5: Comparison of accuracy for targets with referring expressions of different lengths.

### A.4 Hyper-Parameters Setting

Table 5: Performance and efficiency comparison of setting different values of $C_s$.

| Value of $C_s$ | RefCOCOg | | Params | GFLOPs | Inference Time |
| --- | --- | --- | --- | --- | --- |
| | val-u | test-u | | | |
| 64 | 75.72 | 75.16 | 150.75M | 39.91 | 8.66 |
| 128 | 76.11 | 75.67 | 151.24M | 40.33 | 8.72 |
| 256 | 76.98 | 76.54 | 152.44M | 41.16 | 8.80 |
| 384 | 77.10 | 76.50 | 153.94M | 42.01 | 8.99 |

To calculate the correlation map in our Highlighter module, we align the channel dimension of each pixel and word by projecting the visual and linguistic features into a common space. Table 5 shows the effect of different values of $C_s$ on the visual grounding performance and model efficiency. It can be seen that as $C_s$ decreases from 384 to 64, the parameter size, computational overhead and inference time of our framework are consistently reduced. But when $C_s$ exceeds 256, the detection accuracy has only slightly improved (0.12%/-0.04%) on the RefCOCOg dataset. Considering the balance between performance and efficiency, we choose $C_s = 256$ for our Highlighter module.

Similarly, we also conduct ablation experiments to investigate the number of stages $L$ involved in the cascade decoder. Table 6 reports the results obtained by varying the value of $L$ in $[2, 4, 6, 8]$. When $L = 6$, our framework achieves the best performance of 76.98% on the val-u split of RefCOCOg,

---

[1]RefCOCO expressions have an average length of 3.61 while RefCOCO+ have an average length of 3.53, and RefCOCOg contain an average of 8.43 words. (Yu et al., 2016)

Table 6: Performance and efficiency comparison of setting different values of $L$.

| Number of Stages $L$ | RefCOCOg val-u | test-u | Params | GFLOPs | Inference Time |
|---|---|---|---|---|---|
| 2 | 75.80 | 75.47 | 146.14M | 40.92 | 8.58 |
| 4 | 76.33 | 76.11 | 149.29M | 41.04 | 8.71 |
| 6 | 76.98 | 76.54 | 152.44M | 41.16 | 8.80 |
| 8 | 76.27 | 76.02 | 155.60M | 41.28 | 8.88 |

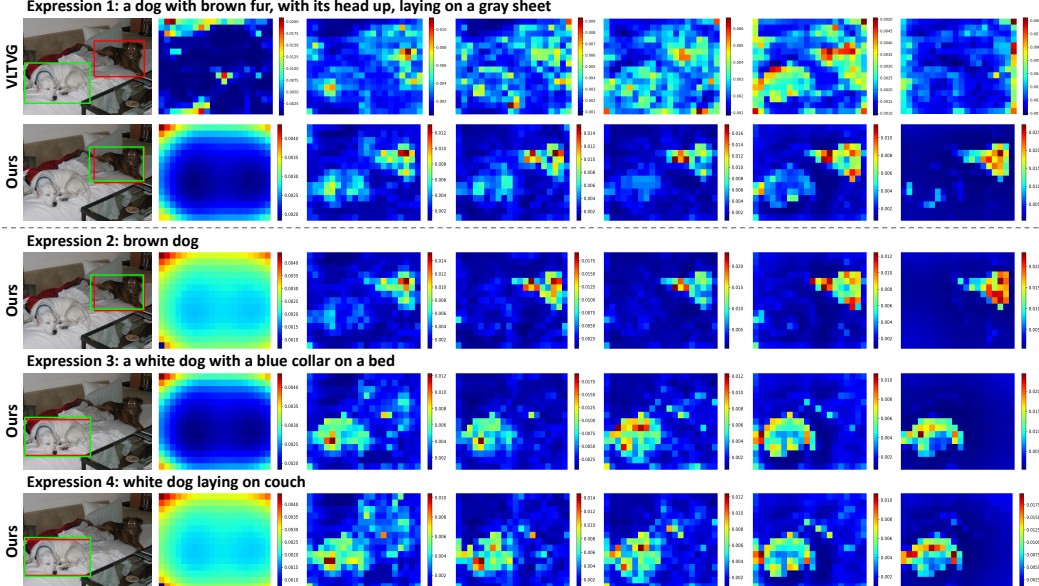

Figure 6: Illustration of the detection results (first column) and the corresponding attention maps generated by different transformer layers (second to seventh columns).

surpassing other settings by 0.65%∼1.18%. On the test-u split, it also obtains the best result 76.54%. Therefore, in our implementation, the cascade decoder contains $L = 6$ stages.

## A.5 DISCUSSION ON THE HIGHLIGHTER MODULE

**Effect on Different Transformer Layers.** This subsection explores the influence of our Highlighter module on the transformer layers in the visual encoder. To this end, we compare the visual grounding results of VLTVG and our framework in Figure 6 (see above the dotted line), and visualize the corresponding attention maps generated by different encoder transformer layers of these two methods. We can observe that VLTVG is not able to effectively align the visual features with the input referring expression, thus producing generic attentions that are widely distributed over the entire image area. In contrast, by integrating Highlighter modules, our method yields more interpretable attention maps that concentrate greater weights on the target object (i.e., "a dog with brown fur"). This finally leads to more accurate detection results compared to VLTVG.

To further validate the above point, in Figure 6, we also present the results and attentions produced by our framework for the same image but with different language expressions (see below the dotted line). It can be found that these attention maps are highly language-dependent, which focus more on the upper right image regions to detect the "brown dog" in expression 2, and pay more attention to the lower left areas for localizing the "white dog" in expression 3 and 4. Moreover, the similar phenomenon can also be observed in Figure 7, where our attention maps are able to capture the "square" and "round" shape information as well as the "partially under" position information associated with the text, thereby accurately detecting different donuts for the two expressions. These experimental

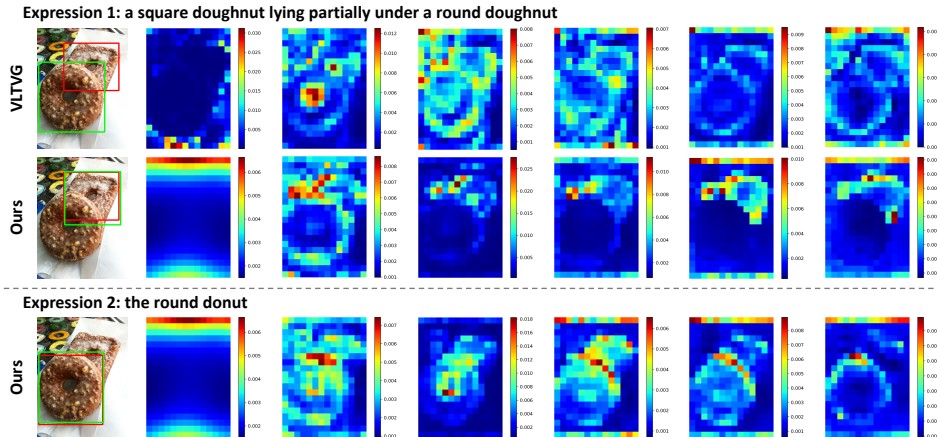

Figure 7: Illustration of the detection results (first column) and the corresponding attention maps generated by different transformer layers (second to seventh columns).

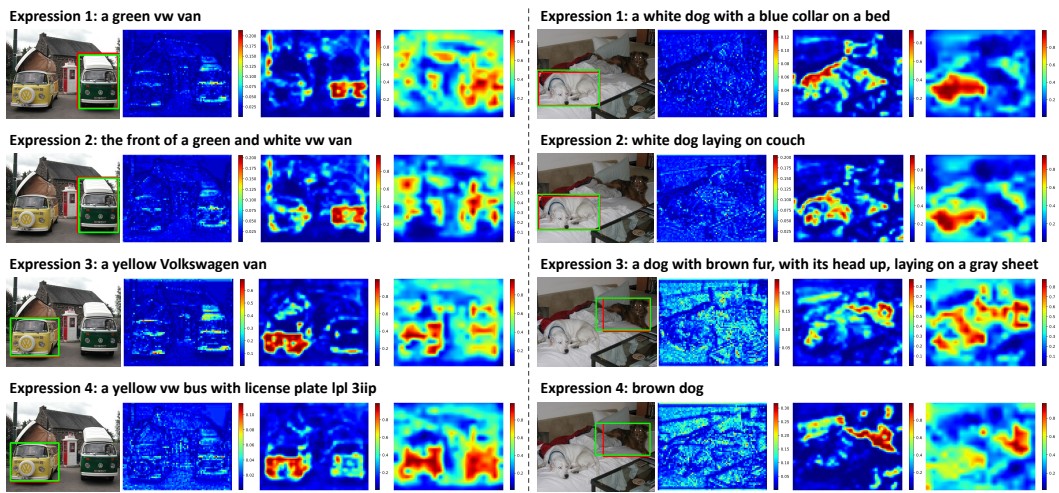

Figure 8: Illustration of the detection results (first column) and the corresponding correlation maps calculated in the Conv3∼5_x residual blocks (second to fourth columns).

results demonstrate the effectiveness of our Highlighter module in indicating the text-relevant visual features.

**Effect on Different ResNet Layers.** Figure 8 showcases the correlation maps calculated by the Highlighter modules in the Conv3_x, Conv4_x and Conv5_x residual blocks of the ResNet. We can see from the results that, since the convolutional operations of different ResNet layers have different receptive fields, the correlation maps in the Conv3_x layer are able to extract some low-level visual information (such as color- and edge-related features), while those in the Conv5_x layer tend to capture more abstract semantic information that is typically associated with the entire objects. Taking the image with two vans as an example, the Conv3_x correlation map of expression 2 highlights more pixels in the "white" areas compared to that of expression 1, and those of expressions 3 and 4 can effectively detect the "yellow" regions on the left van (see the second column in Figure 8). As for the Conv5_x correlation maps, all of them can accurately localize the two "vans" (or "buses") in the image.

In contrast, the correlation map in the intermediate Conv4_x layer comprehensively considers both low-level visual information and high-level concept information in the image, thus capturing more discriminative features located at the image areas around the target referred objects. For example,

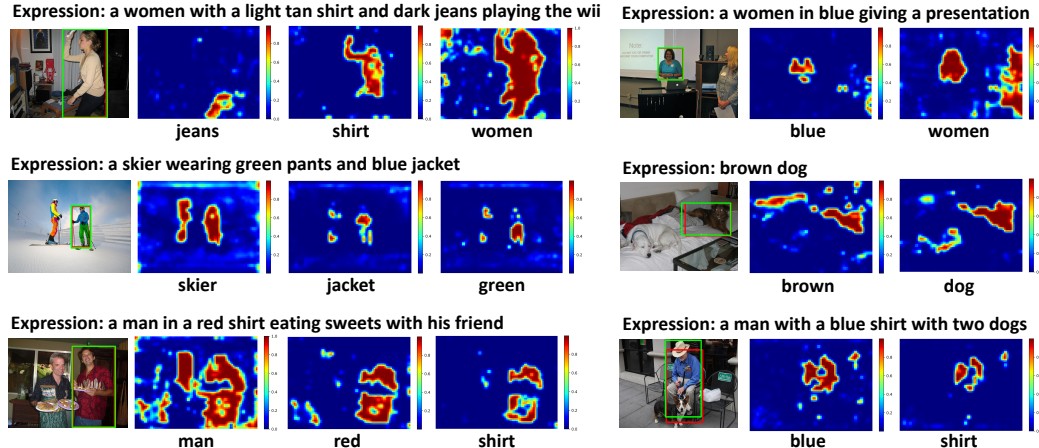

Figure 9: Illustration of the detection results (first column) and the corresponding pixel-wise responses for different words in the Conv4_x correlation maps.

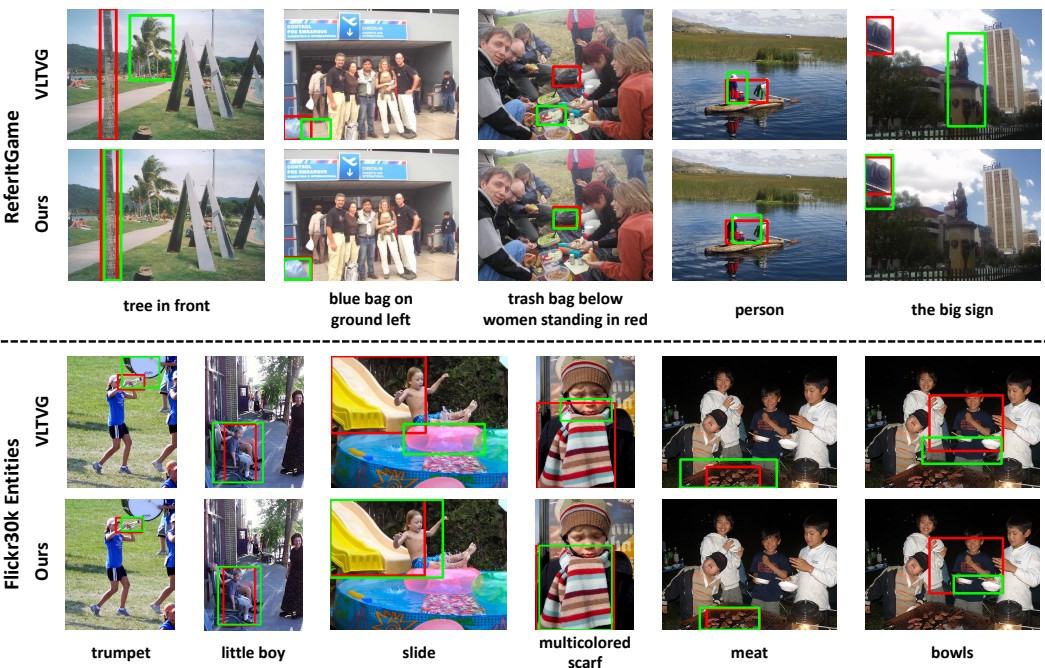

Figure 10: Visualization results of VLTVG and our method on the ReferItGame and Flickr30K Entities datasets. The predictions and ground-truth targets are indicated by the green and red bounding boxes, respectively.

from the Conv4_x correlation maps in the seventh column of Figure 8, we can easily distinguish the "white dog" queried by expressions 1 and 2 from the "brown dog" referred by expressions 3 and 4. All the above results show that our Highlighter module is able to distill useful text-relevant information at different scales.

**Visualization of Pixel-Word Correlation.** Since the proposed Highlighter module calculates the correlation maps at a pixel-word level, in this subsection, we try to investigate how different pixels respond to specific words. For this purpose, we reshape and visualize the pixel responses (i.e., one column in the correlation map $U \in \mathbb{R}^{HW \times N_l}$) for different selected words in Figure 9. It can be observed that most of the feature pixels related to object concepts (e.g., "jeans", "shirt",

"women", etc.) and color attributes (e.g., "red", "green", "blue", etc.) are correctly aligned with their corresponding word. This indicates that our Highlighter module can help to realize a fined-grained visual-linguistic alignment, which brings great benefits for solving the visual grounding task.

## A.6 MORE VISUAL GROUNDING RESULTS

Finally, we qualitatively compare our framework and the previous-best VLTVG method on the ReferItGame and Flickr30K Entities datasets. The detection results displayed in Figure 10 again illustrate the superiority of our method over the state-of-the-art VLTVG, especially on the short expressions in Flickr30K. This may also demonstrate that fully exploiting the pixel-word correlation information by integrating our Highlighter module can effectively promote the final visual grounding accuracy.

