# OpenReview forum: "Improving Visual Grounding with Pixel-Word Correlation and Cross-layer Regularization"
_ICLR.cc/2025/Conference — ICLR 2025 Conference Withdrawn Submission_

### Official Review · Reviewer_i8hF · 2024-10-30

**Soundness:** 2
**Presentation:** 3
**Contribution:** 2
**Rating:** 3
**Confidence:** 5

**Summary:**

The paper focuses on Visual Grounding task. They first provide K-means clustering on the visual features to provide the insight that some of the related works lack discriminative visual features toward the language. Based on the motivation, they come up with a Highlighter module to supervise the correlation map between vision and language, and improve the visual feature. They further involve regularization loss by transferring the bbox to a binary mask as the supervision signal. However, the analysis for the motivation is incomplete, the two ideas are similar with previous works, and the performance is not the state-of-the-art.

**Strengths:**

1. The writing of the motivation is clear.
2. The idea is easy to following.
3. The analysis of figure 1 helps the readers to better understand the motivation.

**Weaknesses:**

1. **The main idea of pixel-word alignment has become very common, therefore resulting in lack of novelty**: the main idea of this paper lies in line 91 to line 95 that the authors want to involve pixel-word level alignment. However, such kind of idea has been explored in previous works, e.g. CoupAlign [1], Albef [2], RIS-DMMT [3], TRIS [4], SegVG [5]. They all have involved cross-attention module between each pixel level feature and each word level feature. And they were published a long time before the date of ICLR 2025 submission, therefore needed to be considered. **The authors should provide a comparison in the effectiveness/efficiency perspective to clarify why it is necessary to involve the proposed Highlighter.** Apart from the quantitative comparison, the motivation in the paper is still common. **Therefore, a further takeaway except for the "pixel-word alignment" should be explored.**

2. **The idea of cross-layer regularization loss has been explored in previous works, therefore resulting in lack of novelty**: This idea transfers the bbox to a binary mask and adopts KL as the loss. However, this approach is highly similar with previous methods. For example, AttBalance [6] transfers the bbox to a binary mask and uses KL to supervise the attention heatmap. They also perform cross-layer supervision; SegVG [5] transfers the bbox to a binary mask as segmentation signal. They also perform cross-layer supervision. **Therefore, the authors need to explain what specific aspect that the proposed loss is novel and if it is better than previous similar approaches.**

3. **Performance is not state-of-the-art**: this paper lacks the comparison with SegVG [5], and the performance is consistently inferior compared with SegVG [5] in a fair comparison.

4. **The scope of training is limited, therefore remains questionable whether the paper has the enough impact to our Visual Grounding field or meets the criteria of ICLR**: in current visual grounding field, people have witnessed the success of large scale training and joint training with other tasks. For example, PolyFormer [7] and MDETR[8] combine the train sets of all the different visual grounding datasets. LLaVA1.5 [9] and Ferrect [10] joint train with VQA and Captiong tasks. However, the authors only train the model on a single training set each time. It remain questionable whether the idea is still effective when scaling up the training scope. **Therefore, the authors should scale up the training to show whether the idea has impact to current field.**

5. **The analysis in Figure 1 lacks comprehensive results**: In figure 1, the authors use K-means to provide analysis and insight that some of the previous works lack discriminative visual features regarding the corresponding text. However, the figure lacks the results of VG-LAW, TransVG++, LUNA, LG-FPN, PVD, which are mentioned and compared in the paper. This analysis is the foundation of the paper to claim that we need to solve the problem of lack of discriminative visual features in line 84 to line 85. **Therefore, a comprehensive analysis toward those method, at least the recent ones, like VG-LAW, SegVG [5] and TransVG++, is necessary.** Otherwise, previous methods might not have such kind of problem. VG-LAW and SegVG [5] have opened their source codes.

[1] CoupAlign: Coupling Word-Pixel with Sentence-Mask Alignments for Referring Image Segmentation, 4 Dec 2022, NeurIPS 2022

[2] Align before Fuse: Vision and Language Representation Learning with Momentum Distillation, 16 Jul 2021, NeurIPS 2021 Spotlight

[3] Beyond One-to-One: Rethinking the Referring Image Segmentation, 26 Aug 2023, ICCV 2023

[4] Referring Image Segmentation Using Text Supervision, 28 Aug 2023, ICCV 2023

[5] SegVG: Transferring Object Bounding Box to Segmentation for Visual Grounding, 3 July, 2024, ECCV 2024

[6] Visual Grounding with attention-driven constraint balancing, 19 Sept 2023, submitted to ICLR 2024

[7] PolyFormer: Referring Image Segmentation as Sequential Polygon Generation, 14 Feb 2023, CVPR 2023

[8] MDETR -- Modulated Detection for End-to-End Multi-Modal Understanding, 26 Apr 2021, ICCV 2021

[9] Improved Baselines with Visual Instruction Tuning, 5 Oct 2023, CVPR 2024 highlight

[10] Ferret: Refer and Ground Anything Anywhere at Any Granularity, 11 Oct 2023, ICLR 2024 Spotlight

**Questions:**

1. Given that LLaVA1.5, Ferrect, Ferrect-v2, MM1.5 have impressive results on visual grounding, what is the meaningful point to still only focus on training on a single set.

2. If the paper focuses on light weight models, why not scaling up the training set to cover all the vg datasets like MDETR and PolyFormer whose model parameters are at the similar range of yours?

---

### Official Review · Reviewer_zCKh · 2024-11-03

**Soundness:** 3
**Presentation:** 2
**Contribution:** 2
**Rating:** 5
**Confidence:** 4

**Summary:**

The paper tackles the problem of visual grounding. Existing methods struggle with either independent feature extraction, leading to redundant visual representations, or complicated language-guided visual encoders that often miss fine-grained visual-linguistic alignment. The authors propose a new module called Highlighter that explicitly calculates pixel-word correlations to enhance visual feature extraction. Additionally, they introduce a cross-layer regularization loss to maintain the consistency of alignment information across different layers. Extensive experiments demonstrate that their approach achieves outstanding performance on multiple benchmarks.

**Strengths:**

- The problem of visual grounding is an important and active field of vision and language field.
- The proposed approach demonstrates good performance on a broad number of downstream tasks.

**Weaknesses:**

- The paper's contributions appear limited in several aspects:
1. Technical Innovation:
    - The layer-wise highlighter injection, which combines cross-attention and word-wise attention, represents only a minor modification to existing attention mechanisms, such as VLTVG's Visual-linguistic verification module and QR-Net's Query-aware Dynamic Attention.
    - Despite the authors' claims that the alignment becomes explicit, the alignment remains implicit as it continues to rely on attention scores.
2. Experimental Limitations:
    - The experiments mainly focus on performance metrics without deeper analysis.
    - For instance, the effects of the correlation loss could have been quantitatively analyzed, particularly in terms of layer-wise correlation map consistency, with or without $L_{cr}$.
3. Writing Structure:
    - The paper suffers from redundant content, particularly in its treatment of previous work.
    - The abstract and introduction contain excessive overlap in their discussion of related research.

**Questions:**

Is the term "cross-layer regularization loss" appropriate for this mechanism? The loss function does not actually consider any interactions between layers, but rather simply computes the sum of individual differences between the mask and correlation values computed independently at each layer. A more precise terminology might better reflect its layer-specific nature.

**Details Of Ethics Concerns:**

There is no concern about ethics for this submission.

---

### Official Review · Reviewer_Rt9F · 2024-11-04

**Soundness:** 2
**Presentation:** 3
**Contribution:** 2
**Rating:** 6
**Confidence:** 2

**Summary:**

The manuscript proposes a novel approach to visual grounding by introducing a “Highlighter” module.
This module leverages pixel-word correlation and cross-layer regularization to enhance visual feature extraction for more accurate language-grounded object localization in images.

**Strengths:**

**Lightweight and Efficient**: The proposed Highlighter module is computationally efficient, introducing minimal overhead while effectively incorporating cross-modality information.
**Backend Compatibility**: The method demonstrates adaptability across various commonly used backbones (e.g., ResNet, Swin Transformer), consistently improving visual grounding performance regardless of the underlying architecture.
**Thorough Ablation**: Extensive ablation studies comprehensively verify the efficacy and robustness of the Highlighter module.

**Weaknesses:**

Lack of Comparison with AttBalance: The proposed method does not include a comparison with AttBalance (Kang et al.), which also employs a mechanism to highlight specific regions in the image for enhanced visual grounding.

[1]: Visual Grounding with Attention-Driven Constraint Balancing; Weitai Kang, Luowei Zhou, Junyi Wu, Changchang Sun, Yan Yan

**Questions:**

See weakness

---

### Note · Authors · 2024-11-13

I have read and agree with the venue's withdrawal policy on behalf of myself and my co-authors.